# BRACTIVATE: DENDRITIC BRANCHING IN MEDICAL IMAGE SEGMENTATION NEURAL ARCHITECTURE SEARCH

## ABSTRACT

Researchers manually compose most neural networks through painstaking experimentation. This process is taxing and explores only a limited subset of possible architecture. Researchers design architectures to address objectives ranging from low space complexity to high accuracy through hours of experimentation. Neural architecture search (NAS) is a thriving field for automatically discovering architectures achieving these same objectives. Addressing these ever-increasing challenges in computing, we take inspiration from the brain because it has the most efficient neuronal wiring of any complex structure; its physiology inspires us to propose Bractivate, a NAS algorithm inspired by neural dendritic branching. An evolutionary algorithm that adds new skip connection combinations to the most active blocks in the network, propagating salient information through the network. We apply our methods to lung x-ray, cell nuclei microscopy, and electron microscopy segmentation tasks to highlight Bractivate's robustness. Moreover, our ablation studies emphasize dendritic branching's necessity: ablating these connections leads to significantly lower model performance. We finally compare our discovered architecture with other state-of-the-art UNet models, highlighting how efficient skip connections allow Bractivate to achieve comparable results with substantially lower space and time complexity, proving how Bractivate balances efficiency with performance. We invite you to work with our code here: https://tinyurl.com/bractivate.

## 1 INTRODUCTION

Researchers manually composing neural networks must juggle multiple goals for their architectures. Architectures must make good decisions; they must be fast, and they should work even with limited computational resources. These goals are challenging to achieve manually, and researchers often spend months attempting to discover the perfect architecture. To overcome these challenges, we turn to the human brain's efficient neural wiring for automated architecture discovery. Neuroscience already underlies core

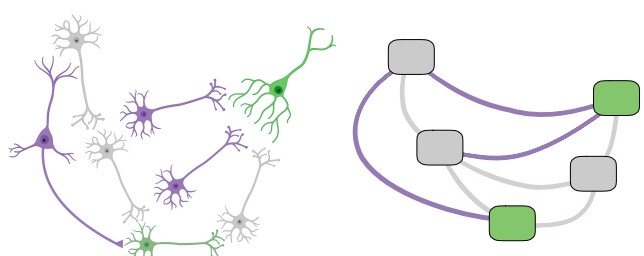

**Figure 1:** Through Bractivate, we discover UNet architecture with high spatio-temporal efficiency by mimicking the brain's dendritic branching.

neural network concepts: The perceptron (Rosenblatt, 1958) is directly analogous to a human neuron. One of the brain's fundamental learning mechanisms is dendritic branching (Greenough & Volkmar, 1973) whereby active neurons send out signals for other neurons to form connections, strengthening signals through that neural pathway. This neuroscience concept inspires us to devise Bractivate, a Neural Architecture Search (NAS) algorithm for learning new efficient UNet architectures networks, capable of being trained twice as fast as the traditional UNet, and often one

to two orders of magnitude lighter in terms of trainable parameters. We apply Bractivate on three medical imaging segmentation problems: cell nuclei, electron microscopy, and chest X-ray lung segmentation.

Medical image segmentation is a growing field in Deep Learning Computer Assisted Detection (CAD): it is a powerful component in clinical decision support tools and has applications in retinal fundus image, lung scan, and mammography analysis. Most papers now approach medical image segmentation with the UNet (Ronneberger et al., 2015); the model architecture is straightforward: it has symmetric, hierarchical convolutional blocks, which are components of an initial contracting path and a final expanding path, with an apex bottleneck layer. Between parallel contracting and expanding blocks, the traditional UNet contains skip connections that pass information through concatenation (Ronneberger et al., 2015). Traditional UNet skip connections involve feature map aggregation with same-scale convolutional blocks, but recent advances have yielded more complex connections ranging from the UNet++ (Zhou et al., 2018) to the NasUNet (Weng et al., 2019). While the UNet is a powerful tool, it does have many limitations:

1. The depth necessary for many segmentation tasks is initially unknown, and traditional neural architecture search (NAS) struggles to identify the optimal UNet depth.

2. Researchers often manually choose skip connection locations, leading to potentially missed optimal connections.

3. Scientists need a NAS algorithm addressing many implementation objectives, including computational time, number of model parameters, and robust segmentation performance.

On a broader level, discovering efficient UNet architectures is crucial because it can generate simpler models for applications on mobile devices, which need low latency for online learning. In the Telemedicine age, many medical applications rely on mobile Deep Learning to segment medical images and process raw patient data (Xu et al., 2017; Vaze et al., 2020). We address the Medical and Engineering fields' need for efficiency with Bractivate, a NAS algorithm to discover lightweight UNet architectures for medical image segmentation tasks. We present the following three primary contributions:

1. An evolutionary algorithm that non-randomly samples from a distribution of various UNet Model depths and skip connection configurations, with both tensor concatenation and addition operators.

2. "Dendritic Branching"-inspired mutations that, just as in the brain, cause salient UNet blocks to branch to other blocks in the network through dendritic skip connections, creating efficient networks that preserve information signals through the network.

3. Bractivate generates high-performing models with lower space complexity than the current state-of-the-art.

The remainder of the paper is structured as follows: In Section 2, we discuss prior works, and what gaps in the literature inspire us to propose Bractivate. Then, in Section 3, we discuss the search algorithm and the dendritic branching mutation. Later, in Section 4, we implement our algorithm with various experiments ranging from changing the search space depth to an ablation study. We report our quantitative and qualitative results, along with baseline comparisons in Section 5 before concluding in Section 6.

## 2 RELATED WORKS

Deep learning algorithms are often restricted to manual model design (Simonyan & Zisserman, 2014; He et al., 2016; Oktay et al., 2018; Ronneberger et al., 2015). To automate model schemes, NAS is the process of selecting candidate architectures through various search strategies to achieve optimal performance (Elsken et al., 2019). Advances in NAS have branched into different areas, including evolutionary algorithms (Miller et al., 1989; de Garis, 1990; Yao, 1993; Fogel et al., 1990; Angeline et al., 1994; Real et al., 2018; Yao, 1999) and automatic pattern recognition (Cai et al., 2018; Radosavovic et al., 2020). While both approaches are merited, the tasks address image classification problems, and although some focus on skip connections, they lack deeper investigation

of their optimal configurations. Recent advances in the UNet have led to alternative skip connection implementations, including addition (Ghamdi et al., 2020), max out operations (Estrada et al., 2019; Goodfellow et al., 2013) and multiplication by a gating function (Oktay et al., 2018). Ghamdi et al. (2020) reports these connections' improved efficacy over traditional concatenation, as they overcome vanishing gradients and preserve salient features.

Auto-DeepLab, which Liu et al. (2019) present for semantic segmentation, is a graph-based NAS algorithm that addresses changing model depth and connection locations in hierarchical models. Building off this work, Zhou et al. (2020) propose a similar graph-search algorithm, termed UNet++, for improved NAS; the final model incorporates dense skip connections to achieve multi-scale feature aggregation. Although UNet++ successfully addresses the model depth problem, it ignores choosing the skip connection operator and relies on pretraining and pruning to generate skip connection configurations.

The Differential Architecture Search (DARTs) algorithm by Liu et al. (2018) continuously relaxes the architecture representation to enable gradient-based optimization. Advancing this algorithm, Chen et al. (2019) proposes the Progressive Differentiable Architecture Search Algorithm (PDARTs) to allow the searched model's depth to grow during the search; when applied to ImageNet (Deng et al., 2009), CIFAR-10 (Krizhevsky et al., 2009), or CIFAR-100 (Krizhevsky et al., 2009), the total training time is approximately seven hours.

Although the DARTS and PDARTs algorithms are specific to image classification and sequential model architecture, they lack applications for segmentation models. Weng et al. (2019) suggest a NASUNet method with modified DARTs search for medical imaging segmentation; their approach addresses searching for model parameters in the convolutional blocks to reduce the space complexity found in attention-based (Oktay et al., 2018; Hy, 2018) and recurrent (Alom et al., 2018; Hy, 2018) UNets, yet NASUNet still preserves same-scale concatenation skip connections, overlooking alternate skip connection possibilities across network blocks.

Many existing NAS algorithms use modified objective functions for evaluating the searched model performance *e.g.* NAS-Bench-101 (Ying et al., 2019) uses the cross-entropy loss, Stochastic Neural Architecture Search (SNAS) (Xie et al., 2019) devises a cost function deemed Memory Access Cost (MAC) that incorporates the floating-point operations (FLOPs) and number of parameters, and PDARTs (Chen et al., 2019) employs auxiliary loss (Szegedy et al., 2014). To target gaps in the literature related to skip connection search for efficient models, we propose Bractivate, a NAS algorithm inspired by the brain's dendritic branching to facilitate optimal architecture discovery.

## 3 THE BRACTIVATE NAS ALGORITHM

### 3.1 DENDRITIC ARBORIZATION

Table 1 translates neuroscience into computational terms we use throughout the paper. In the neuroscience field, dendritic branching occurs when stimulating environments cause neurons to form new connections (Greenough & Volkmar, 1973; Greenough et al., 1985). These neural connections are associated with learning, and even learning-impaired children with fetal alcohol syndrome display lower dendritic branching levels (Hamilton et al., 2010) compared to their healthy peers. This branching phenomenon parallels Deep Neural Networks: in the brain, dendrites form new connections to the hyperactive soma: the perceptron's activation function is

**Table 1:** Defining terms from neuroscience we use and their parallels in Deep Learning.

| Neuroscience | Deep Learning |
| --- | --- |
| Soma | Layer in a block |
| Dendrite | Input Connection |
| Axon | Sender output Connection |
| Dendritic Branch | Connection to first layer in an active receiving block |

to the biological soma as the incoming connections are to dendrites. Perceptrons can be stacked together to form multi-layer perceptrons (Rumelhart et al., 1986), with parallel architecture similar to the brain, and this structure underlies convolutional neural networks (LeCun et al., 1995).

For the UNet, if we consider layer in the network's blocks to be a neural soma, then we can think about a block's "activity" as the mean-absolute value of its layers' activations, as shown by Equa-

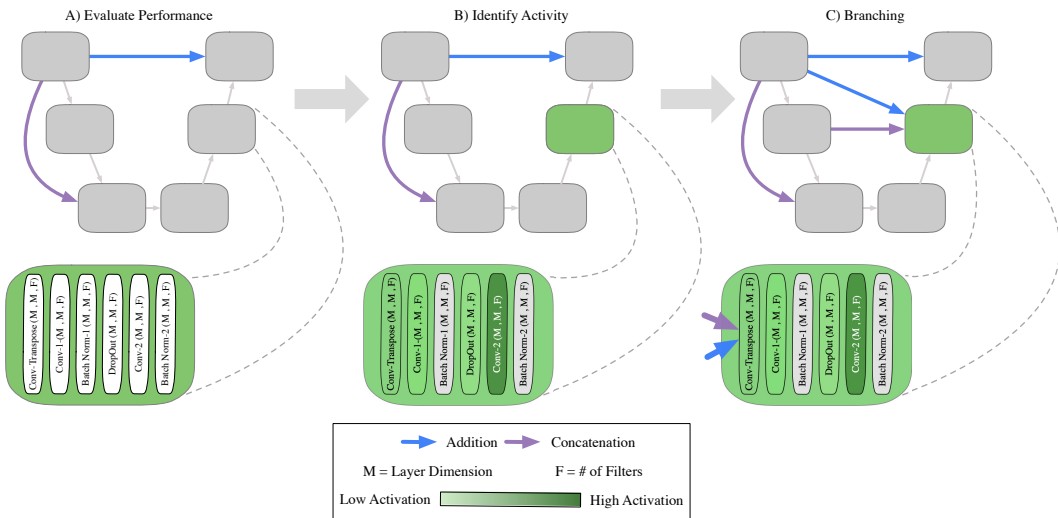

**Figure 2:** During each search iteration, Bractivate chooses a model from the randomly initialized search domain. After we train on the dataset, we evaluate its performance. To mutate, we identify the most active block in the "best model" architecture, as per Equation 1; to that active block, we initialize new skip connections (branches) pointing from other blocks in the network to the highest-activation block, propagating their signals through randomly chosen branches.

tion 1.

$$A_b = \frac{1}{L} \sum_{l=0}^{L} |\mathbf{A}_l| \tag{1}$$

where $A_b$ represents block activation, $b \in B$, $\mathbf{A}_l$ is the weight of each layer in the block, $l$, and $L$ is the total number of layers in the block. Knowing the block's location, $b$, with $\max(A_b)$, surrounding blocks then form skip connections around this active node, a process analogous to dendritic branching. We apply this method to target conv and deconv layers, excluding batch normalization layers as they contain static weights and high values that overwhelm the mutation's layer selection. When new connections are formed, across blocks with various tensor dimensions, we overcome spatial dimensional mismatch by resizing the incoming connection tensors to the receiving tensor by bilinear interpolation.

## 3.2 NAS with Dendritic Branching Mutations

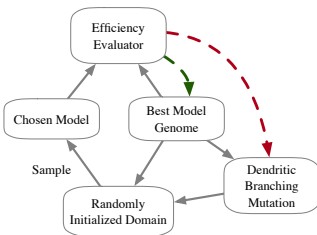

**Figure 3:** Bractivate samples from a randomly initialized domain, constrained by the model depth, $D$. The efficiency evaluator compares a selected model with the current "best model" genome. If the "best model" outperforms the current model, we mutate the "best model" (Figure 2) and replace the chosen model with the mutated version in the search space.

We implement our dendritic branching mutations to our evolutionary algorithm based on adding dendritic branching mutations to a randomized search domain, as displayed in Figure 3. This mutation applies a mutation operator to form new skip connection combinations to the most active block in the network. This NAS algorithm initializes a random domain with $n$ model genotype. Each genotype codes for the model's depth, $D$. It also encodes the number of filters per conv and deconv layer, the degree of skip connections for each block, and the skip connection operator type (concatenation or addition). A detailed discussion on the genotype initialization and its micro-architecture is found in Appendix A.1. When

initializing the genotype, we constrain the number of feature maps to grow by 1.5 in the encoder and decrease it by the same factor for each block in the decoder.

## 3.3 EFFICIENT LOSS

NAS methods often focus on accuracy as the main performance metric (Real et al., 2018; Radosavovic et al., 2020), but often lack consideration for discovered model space and time complexity. To address this, we propose an efficient loss function for the NAS evaluation step. Traditional binary cross-entropy is given by Equation 2. During the NAS mutation and selections, the search process accelerates as "better" models have faster training steps.

$$BCL = -\frac{1}{m}\sum_{i=1}^{m}(\mathbf{y}_i \times \log(\hat{\mathbf{y}}_i) + (1 - \mathbf{y}_i) \times \log(1 - \hat{\mathbf{y}}_i)) \tag{2}$$

where $m$ is the number of samples, $\mathbf{y}_i$ is the sample image's true segmentation mask tensor, and $\hat{\mathbf{y}}_i$ is the model's prediction tensor. We propose an alternate efficient loss function equation, *Efficient Loss Scaling* ($EL_S$). It uses the number of model parameters, $P$, and the training time per epoch, $T$.

### EFFICIENCY LOSS SCALING

We also explore Efficiency penalty scaling where $\log(P)$ and $\log(T)$ scale the overall loss function through multiplication, hence:

$$EL_S = \gamma \times \log(P) \times \log(T) \times BCL \tag{3}$$

In our experiments we set $\gamma = 0.01$. We use Equation 3 in Section 4.4 during model search. A detailed ablation study on how this equation favors efficient networks and outperforms standard $BCL$ can be found in Appendix A.3.

## 4 EXPERIMENTS

### 4.1 IMPLEMENTATION

For the following experiments, we execute our programs using TensorFlow 2.3.0 and Keras 2.4.3, using Python 3.6.9. For training the models and inference, we use a virtual machine with 25 GB of RAM, 150 GB of disk space, and one Tesla v-100 GPU.

### 4.2 THE DATA

To validate Bractivate NAS, we work with three datasets:

1. The Montgomery Pulmonary Chest X-Ray (Jaeger et al., 2014) dataset (N = 138)
2. The EPFL CVlab Electron Microscopy (Lucchi et al., 2011) dataset (N=330)
3. The Kaggle Data Science Bowl Cell Nuclei Segmentation Challenge (Caicedo et al., 2019) dataset (N=670)

A detailed discussion of the dataset follows in Section 4.3.

### 4.3 PREPROCESSING

We resize all images in the three datasets to be 128×128 pixels with bilinear interpolation, and later min-max normalize the pixels, so that pixel intensities are $\in [0, 1]$. We apply the same preprocessing to the image masks. We use a 0.1 validation split percentage on all three datasets to evaluate the optimized model loss function and the Dice coefficient during the search.

### 4.4 PENALIZING TIME AND SPACE COMPLEXITY

We experiment with using the standard binary $BCL$ loss function in Equation 2, and the $EL_S$ function in Equation 3. We record our results both with the Dice Score, representing the precision and recall harmonic mean. We note that these loss functions are used only during the search step. During full model training, we use the $BCL$ loss on the "best model" and use that for our later experiments in Sections 4.7 and 4.8.

### 4.5 DISCOVERING ARCHITECTURE

For all three datasets, we use the same Bractivate search algorithm based on dendritic branching. We initialize our search domain as a queue with 20 randomly generate model genotypes and initialized generated models from the genotypes; we then train the models for 50 epochs on the datasets with Early Stopping. The Early Stopping has patience of 10 and $\min \Delta$ of 0.05. In a first-in-first-out (FIFO) fashion, we evaluate each genotype in the queue: if the generated model has $EL_{\min}$, then it becomes the "Best Model," and we place this genotype back in the queue. Suppose it has a $EL > EL_{\min}$. In that case, the search algorithm mutates the "Best Model" genotype with the dendritic branching method described in Figure 2 before replacing the mutated genotype in the search queue.

### 4.6 GOING DEEPER

We notice that the original Bractivate search algorithm with a minimum depth of two yields mainly shallow networks, usually with two or three expanding and contracting blocks. To explore how model depth affects the search algorithm's results, we constrain the search space such that depth is $\in [5, 10]$ and later $\in [7, 10]$, and observe how the Dice coefficient and Dice:time ratio change while using the $EL_S$ (Equation 3).

### 4.7 ABLATING BRANCHING CONNECTIONS

We hypothesize that active block branching increases signal propagation, reducing time complexity, and improving overall model performance. Thus, we must prove these new branches are necessary for the model's success by ablating them and measuring how their absence affects the model performance. We perform these experiments by training the selected model architecture for 200 epochs on the dataset, and use the Keract (Remy, 2018) library to measure layer activations on the most active layer (the second convolutional layer in the decoder of a $D = 5$ deep UNet. For each layer, we calculate the layer's average activation from Equation 1 and then ablate all dendritic (input) connections of the most active block. We record the results quantitatively with the Dice coefficient and visualize them by analyzing changes in the activation heat maps.

### 4.8 BASELINE COMPARISON

We compare Bractivate models to other state-of-the-art methods, including the standard UNet model (Ronneberger et al., 2015), Wide-UNet (Zhou et al., 2018), UNet++ (Zhou et al., 2020), and attention-based models . We obtain all model architectures from GitHub repositories made by these models' authors or by contributors who create implementations in Keras and use them with Xavier weight initialization on the three datasets for comparison.

## 5 RESULTS

We observe clear patterns pointing to how dendritic branching allows for efficient neural network selection through our experimentation. Figure 4 presents an example of the discovered architecture with the $ELS$ function.

### 5.1 DISCOVERED ARCHITECTURES

Because the mutations add more connections to salient blocks, the most active node has the highest input connections. Bractivate randomly assigns addition or concatenation connections to the most active block, optimizing the best connecting operator combination.

Figure 4 shows the discovered architecture when the search space is constrained such that $D \in [5, 10]$. Al Ghamdi *et al.* (Ghamdi et al., 2020) report that the addition operator can perform as well or better than the concatenation operator, so we incorporate both operators as modules into the NAS, with each mutation yielding a new random connecting operator combination to the salient block.

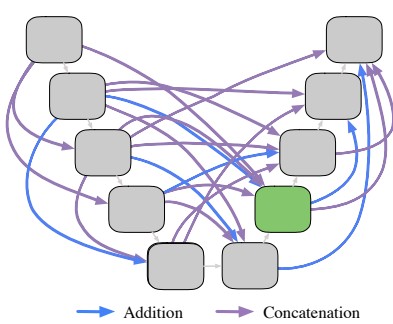

**Figure 4:** A sample of the discovered architecture through the efficiency loss scaling in Equation 3. The green block is the most active block, having the highest number of incoming dendritic connections.

## 5.2 DEPTH AND TIME-SPACE COMPLEXITY

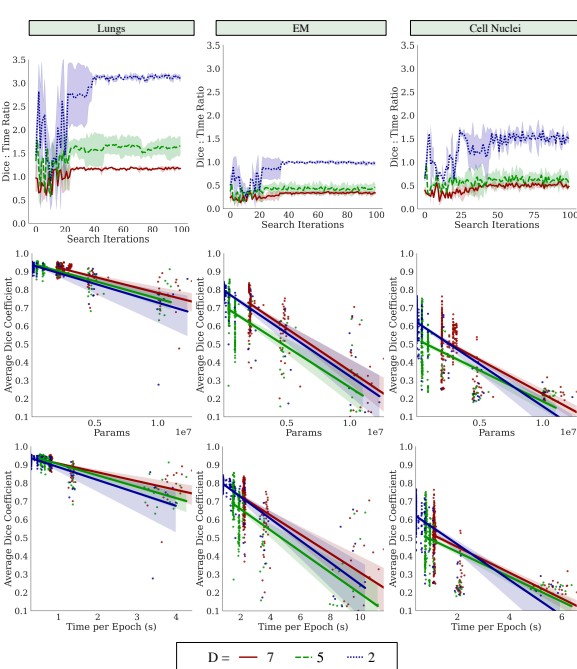

In Figure 5, we observe a negative correlation between the training time per epoch and the Dice coefficient, indicating that Bractivate favors shallower over deeper networks, as the skip connections branching towards active blocks carry the necessary information for the segmentation task. This slope varies between datasets; a simple problem like segmenting two-component lungs confers an overall larger Dice: time ratio than the cell nuclei and electron microscopy tasks. These problems are challenging because they have more than two connected components, yielding lower Dice: time ratios. More importantly, our methods determine that given the early stopping constraint, shallower models ($D = 2$) with fewer parameters have comparable performance to deeper models. When UNet models are too deep ($D \in [7, 10]$), the input signal may be lost in transformations, leading to lower performance; shallower models preserve this signal.

**Figure 5:** Comparing the effects of model depth on the correlations between the Dice coefficient performance metric and different depth constraint $D$ over three trials. Top: Dice vs time per epoch; Middle: Dice vs. number of parameters (spatial complexity) Bottom: Dice vs the number of model params.

## 5.3 SKIP CONNECTION ABLATION STUDY

Our ablation study most strongly confirms our hypothesis that dendritic branching to active blocks significantly improves segmentation performance: Figure 6 examines the saliency maps produced by the model architecture in Figure 4 before and after ablating connections to the most active block. Before ablation, the most active block's saliency maps show encoded information that significantly influences the decoded feature maps during deconvolution.

After ablation, the saliency maps for the EM and nuclei segmentation tasks lack accurate late-stage saliency maps. When the salient block has configured dendritic branches from neighboring blocks,

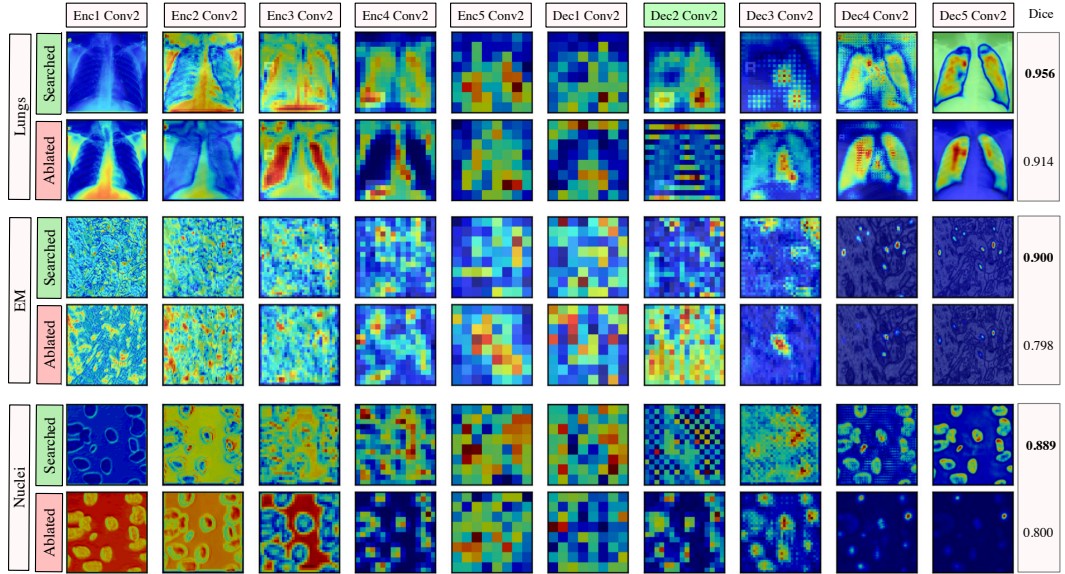

**Figure 6:** Activation map comparison for the second deconv block's second convolutional layer for the architecture described in Figure 4. Blue represents low saliency, and red represents high saliency. We ablate the most salient block (The green block containing Decode2 Conv2) in the network and display activation differences after ablation.

the output signal is highly accurate. However, when these vital encodings in the Decode 2 block lack input from neighboring blocks, the output signal is degraded. This degradation is especially true for the EM and nuclei segmentation tasks.

The EM and nuclei segmentation tasks contain more than two connected components; removing dendrites to salient blocks prevents valuable information from neighboring blocks to travel to the most salient block, degrading the signal through the last four blocks in the network. The model's Dice score is significantly lower in the ablated architecture than in the intact Bractivate-selected model. The added information from these dendritic skip connections, explicitly targeting a salient block in the model, generates more accurate saliency maps, helping the model learn faster. Before ablation, activations are more salient during the decoding phase than post-ablation, where saliency concentrates in the encoder. This observation may be because removing connections towards an active block forces surrounding layers to compensate by increasing their activations.

## 5.4 BASELINE COMPARISON

**Table 2:** Comparing the spacial complexity of Bractivate with various state-of-the-art UNet architectures. The top group represents manually-designed models. The middle row comprises differentiable search. The bottom is ours, based on dendritic branching. We report the results as "Dice (# Params)."

| Model | ‖ | Lung | EM | Nuc |
|---|---|---|---|---|
| UNet (Ronneberger et al., 2015) | ‖ | 0.925 (7.7e6) | 0.811 (7.7e6) | 0.833 (7.7e6) |
| R2-UNet (Alom et al., 2018) | | 0.596 (9.5e7) | 0.464 (9.5e7) | 0.049 (9.5e7) |
| Attn-UNet (Oktay et al., 2018) | | **0.954** (3.2e7) | **0.937** (3.2e7) | 0.721 (3.2e7) |
| UNet++ (Zhou et al., 2018) | | 0.903 (9.0e6) | 0.846 (9.0e6) | 0.841 (9.0e6) |
| WideUNet (Zhou et al., 2018) | ‖ | 0.888 (9.3e6) | 0.811 (9.3e6) | 0.828 (9.3e6) |
| NasUNet (Weng et al., 2019) | ‖ | 0.934 (1.2e5) | 0.729 (4.8e5) | 0.774 (**1.2e5**) |
| **Bractivate** | ‖ | **0.942** (**3.1e4**) | **0.929** (**4.8e5**) | **0.878** (**4.8e5**) |

Figure 7 highlights how Bractivate achieves comparable performance to larger models when initialized with Xavier initialization (Glorot & Bengio, 2010). Table 2 highlights how Bractivate is significantly smaller than many of the other state-of-the-art models: it exchanges high spatial com-

plexity for more skip connections, as these branches allow information to propagate through salient blocks in the network. For domain-specific tasks, high parameters reduce the signal: noise ratio in the network; simpler models like Bractivate rely on powerful skip connections, analogous to dendrites, to carry most of the signal. Because these connections consist of simple concatenation or addition operators, they greatly reduce the number of trainable parameters, preventing overfitting; this speaks to Bractivate's comparable–or better–Dice scores as compared to the baseline models.

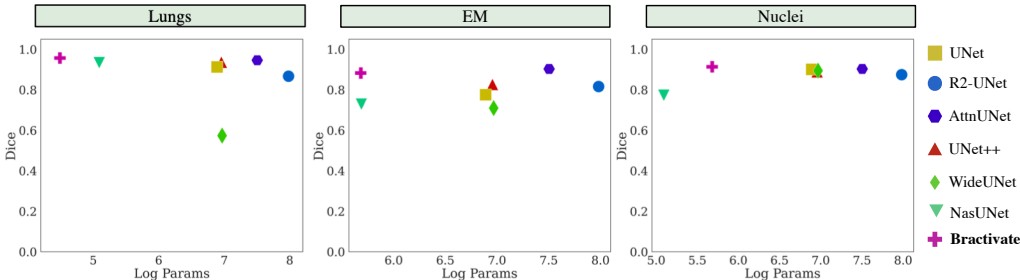

**Figure 7:** Comparing model performance measured by the Dice score as a function of different model architectures. Note that these models are all Xavier initialized.

## 6 CONCLUSION

Throughout this paper, we highlight how dendritic branching in the brain inspires efficient skip connections in Deep Learning models. With our focus on segmentation, we present Bractivate as a method for identifying skip connection configurations to elevate the traditional UNet. During the search, Bractivate mutates the architecture so that the most salient blocks in the network branch out their "dendrites" to other network blocks. By replacing the oldest model in the search space with the new mutated architecture, we accelerate the search rate.

The ablation study strongly supports our hypothesis that dendritic branching is necessary for efficient model discovery; when we ablate dendritic connections to the most salient block, the Dice Score decreases. Before and after ablation, the saliency maps reveal stark contrasts, with the ablated activation maps lacking apparent features for the final segmentation layer in the UNet's decoder. We finally weigh our methods with other baselines, highlighting how smaller networks can perform segmentation tasks well given limited pretraining data.

Overall, we present how optimally configured skip connections, inspired by the brain, yield robust signal streaming paths through a lightweight network. Our algorithm is an asset to many mobile medical computing technologies that rely on low latency and high computational efficiency.

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

## A  APPENDIX

### A.1  GENOME MICRO-ARCHITECTURE

When designing our search space, we formulate genotypes that code for model architectures. Following common patters in convolutional networks, and the UNet Ronneberger et al. (2015), we first impose the following constraints on our search space:

- The network blocks must be symmetrical. This means that the number of blocks both in the network encoder and decoder are identical, with mirror internal layer configurations (types of layers, numbers of filters, and number of layers in the block)

- The network must be hierarchical. When designing models for medical image segmentation, we rely on hierarchical backbones for both the encoder and decoder, as reflected in Figure 4.

- We constrain skip connection directionality. In the network, skip connections only occur from earlier to later layers in the background.

Figure 8 shows the standard micro-architecture for the contracting and expanding parts of the network. We also note that while the layer arrangements are constant, the number of filters, $n$, for each block is initially random. However, each integer value of filter numbers is scaled by a factor of 1.5 for each subsequent block, as Figure 8 highlights.

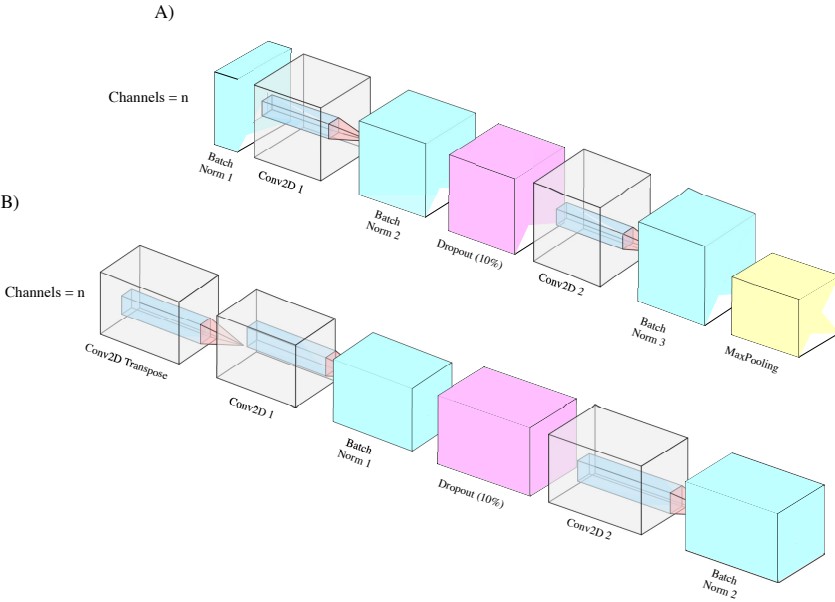

**Figure 8:** Each prism represents a layer in the overall repeating block motifs in the network. A) The contracting block micro-architecture for one bloc. Note that this motif repeats throughout the contracting phase of the network. $n$, the number of channels, is factored by 1.5 for each subsequent contracting block. B) The Expanding block of the micro-architecture. $n$, the number of channels, is factored by .75 for each subsequent expanding block.

## A.2 GPU RUN-TIME

Overall, the search algorithm had GPU run-times, as shown in Table 3. We note that these results are reported after running the search algorithm on a Tesla-v100. The reported values are an average of three independent trials. Oftentimes, the run time was dependent on the dataset size. Because the Cell Nuclei dataset had the highest number of sample images, it took longer to train on as compared to the smaller Lung dataset.

**Table 3:** GPU Run-time for all three datasets with time measured in hours averaged over three trials with image dimensions of $128 \times 128$.

| Dataset | Run Time (hrs) |
|---|---|
| Lungs | $0.483 \pm 0.067$ |
| EM | $0.878 \pm 0.164$ |
| Cell Nuclei | $1.31 \pm. 0.170$ |

## A.3 ABLATING EFFICIENT LOSS

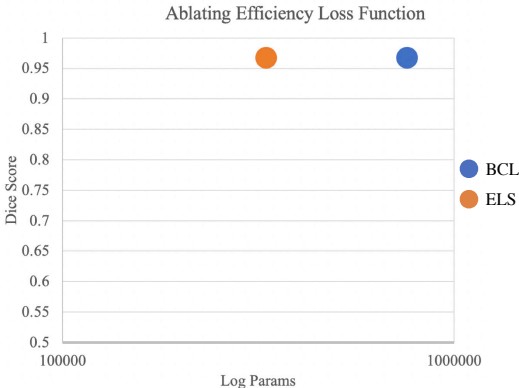

**Figure 9:** The efficiency loss scaling (ELS) loss function selects a smaller model (orange) that can perform at the level as one selected simply by BCL alone (blue).

We also examine the effect of ablating the efficiency loss' parameter and time penalty terms on the overall model selection. Through our investigation, we find that the efficiency loss does help the model select smaller models, that can perform at the level of larger models selected by the BCL loss function. Figure 9 highlights this trend for the Lung Dataset. The results are averaged over three trial runs.

We see that removing the penalty terms for high space and time complexity still yields high-performing models. However, these models are larger, and computationally costly.

