# OpenReview forum: "Bractivate: Dendritic Branching in Medical Image Segmentation Neural Architecture Search"
_ICLR.cc/2021/Conference — Reject_

### Official Review · AnonReviewer2 · 2020-10-25
**Unclear contribution and insufficient empirical comparison**

**Rating:** 3
**Confidence:** 3

**Review:**

The authors propose a neural architecture search (NAS) algorithm inspired by brain physiology. In particular, they propose a NAS algorithm based on neural dendritic branching, and apply it to three different segmentation tasks (namely cell nuclei, electron microscopy, and chest X-ray lung segmentation). The authors share their codes with the scientific community, which is highly appreciated.

- I have a concern about one of the main motivations of the paper. The authors say that they "take inspiration from the brain because it has the most efficient neuronal wiring of any complex structure" (lines 7 and 8 in the Abstract). My question is: What is the problem with using biologically implausible NAS methods or with the fact that some approaches are incompatible with current understandings in neuro-biology? I consider that all proposals to improve neural networks (NNs) training and design are welcome, but I do not see the relevance of explicitly searching for biologically realistic algorithms in NNs. As Yann LeCun would say (http://matt.colorado.edu/compcogworkshop/talks/lecun.pdf): "Let's be inspired by nature, but not too much". It is indeed nice to imitate nature, but we mainly need to understand what is actually relevant for our practical purposes. For instance, as pointed out by LeCun, in the case of airplanes, we developed aerodynamics and compressible fluid dynamics, and we figured out that feathers and wing flapping were not crucial. In this sense, I'd like to see a better justification and contextualization of this dendritic branching approach.

- The paper, in my humble opinion, is a bit confusing, because the different contributions and overall structure of the paper dilute the main focus of the paper. The paper first starts with the presentation of a novel bioinspired NAS method. Then, we move towards medical imaging segmentation and the limitations of U-Net, and later we discover that there are also evolutionary computation algorithms involved (as a primary contributions that is not even cited in the Abstract). I would encourage the authors to rewrite the paper in a clearer way, identifying the core contribution and challenges, as well as the motivation and the rationale behind the approach they use. In particular, I'd like to better understand how the submitted paper compares (quantitatively and qualitatively) to prior art, and what are the conceptual or empirical advantages of the proposed approach.

- Why other NAS algorithms were not included in the experimental comparison (apart from Weng et al. (2019))? Regarding the competitor methods, why not to introduce in the comparison methods like Neuron-Evolution for Augmenting Topology (NEAT) [1], HyperNEAT [2], ENAS [3] or [4], that represent popular methods in neural architecture search? Without a more extensive experimental comparison with prior methods and different strategies is difficult to elucidate the actual empirical contribution of the proposed method.
[1] Kenneth O. Stanley, and Risto Miikkulainen. "Evolving neural networks through augmenting topologies." Evolutionary computation 10(2): 99-127, 2002.
[2] Kenneth O. Stanley, David B D’Ambrosio, and Jason Gauci. "A hypercube-based encoding for evolving large-scale neural networks". Artificial life, 15(2):185–212, 2009.
[3] Pham, Hieu, Melody Y. Guan, Barret Zoph, Quoc V. Le, and Jeff Dean. "Efficient neural architecture search via parameter sharing." arXiv preprint arXiv:1802.03268 (2018).
[4] Zoph, Barret, and Quoc V. Le. "Neural architecture search with reinforcement learning." arXiv preprint arXiv:1611.01578 (2016).

- Evolutionary computation techniques have been employed many times for both training neural networks and or designing their architecture and building blocks (e.g., activation functions), with papers on the subject already published in the 80's and 90's. From this point of view, statements like "Within the last five years, advances in NAS have branched into different areas, including evolutionary algorithms (Real et al., 2018)" look insufficient. The related works section is, in my opinion, not complete and clear enough. See the following papers as example:
[5] Miller, Geoffrey F., Peter M. Todd, and Shailesh U. Hegde. "Designing Neural Networks using Genetic Algorithms." ICGA. Vol. 89. 1989.
[6] de Garis, H. (1990). "Genetic programming: Modular evolution for Darwin machines". In Proceedings of the 1990 International Joint Conference on Neural Networks (pp. 194-197).
[7] Fogel, David B., Lawrence J. Fogel, and V. W. Porto. "Evolving neural networks." Biological cybernetics 63.6 (1990): 487-493.
[8] Yao, X. (1993). "Evolutionary artificial neural networks". International journal of neural systems, 4(03), 203-222.
[9] Angeline, Peter J., Gregory M. Saunders, and Jordan B. Pollack. "An evolutionary algorithm that constructs recurrent neural networks." IEEE transactions on Neural Networks 5.1 (1994): 54-65.
[10] Yao, X. (1999). "Evolving artificial neural networks". Proceedings of the IEEE, 87(9), 1423-1447.

- In relation to the proposed model itself (called Bractivate), there are many aspects that remain unclear:
(1) What do the authors mean by semi-random evolutionary algorithm?
(2) What evolutionary algorithm do they use? It appears that they just apply a mutation operator on the most active block in the best architecture found so far.
(3) The authors propose a loss that includes the number of model parameters and the training time per epoch. In this sense, what do the authors mean by BCE in Equation 3? I guess they actually refer to BCL, right?
(4) If I understood correctly, Bractivate does not allow to create or insert new blocks but it only branches already existing blocks (using two skip connection operator types: concatenation or addition). This seems a severe limitation in the neural architectures it can generate. Am I correct?
(5) The authors state that "Our ablation study most strongly confirms our hypothesis that dendritic branching to active blocks significantly improves segmentation performance". Did the authors employ several independent runs, some experimental validation protocol (like Cross-validation) and statistical tests to verify the existence of statistically significant differences between the methods under comparison?

---

### Official Review · AnonReviewer1 · 2020-10-28
**The paper tackles the problem of image segmentation leveraging neural architecture search. Inspired by dendritic branching, the proposed method, termed Bractivate, identifies the most salient block in sampled candidate and mutates the architecture by randomly generating connection from other blocks. The results show the improvement of the segmentation on several datasets.**

**Rating:** 4
**Confidence:** 4

**Review:**

The goal of the proposed Bractivate is to search an architecture from the predefined search space which performs well and efficient, as the proposed efficiency loss simultaneously constrains the parameter and training cost of candidate. However, the proposed method misses many crucial details, which may make the empirical results not convincing to me.

My major concerns are as follows:
1) The paper did not provide the micro-architecture of model genotypes, so the initialized search domain is ambiguous. As I understand, the micro-architecture of U-Net’s block is fixed and only searching for the connections between blocks.
2) The article did not give experimental results or explanations to prove the rationality of the formulation in Eq. 1. Why the block with max (Ab) should be chosen? And what the difference if the algorithm chooses another block?
3) The GPU hours it takes to finally search an optimal architecture. Evolutionary algorithms are usually time-consuming so decreasing the necessary of searching compared with other manual-tuned baselines.
4) How to deal with the miss-matching of spatial dimension and channels when aggerating the activations of different blocks?
5) What about applying the proposed Bractivate into other popular segmentation benchmarks and see if the proposed method can still achieve SOTA performance, e.g., COCO, Pascal etc.

Besides, there’s a strict upper limit of **8** pages for the main text of the initial submission this year. However, this submission breaks this rule and the conclusion part exceeds 8 pages.

---

### Official Review · AnonReviewer3 · 2020-10-28
**This paper proposed a methods inspired from the brain to add skip connections on the UNet, which can improve the network’s performance. At each iteration, the model adds a new skip-connect branch to the most active block. Experiment results on some datasets are shown.**

**Rating:** 4
**Confidence:** 4

**Review:**

Pros:
1.	Borrow neuroscience concepts into NAS problem is novel.
2.	The paper claims that healthy brain has more branches, thus adding new skip-connections can improve the performance. This idea is of great interest.

Cons:
1.	Your model is too simple so that the technical contribution is too limited. You claim that the healthy have more dendritic branches in their brain, but the differences between their somas are not discussed. Thus, it may enough to only add skip connections on a network to improve its performance. The layers may also need to be tuned.
2.	It is not clear that how many skip-connections should be added to the network. Do more edges lead to higher performance? If so, why do not we directly use densely connected network?
3.	The experiment results show your model cannot reach SOTA in 2/3 datasets.
4.	No ablation study of the proposed efficient loss.
5.	I suggest you improve your set type. It is not easy to detect the main text and the caption of Figure 3 and Figure 5.
6.	In Figure 6. You should explain the meaning of the color. In caption, you should explain the effectiveness with more detail. Such as which part of the image is highlight and what does it mean.
7.	You need to improve you writing. There are too many strange expressions in your paper. In Sec 3.3 you can say “NAS methods often focus on accuracy, which is regarded as the main performance metric”. In Sec 4.4 you can say “We experiment with the standard binary BCL loss function in Equation 2, and the ELs function in Equation 3.” You may need lots of efforts working on this.

Overall Review:
This paper borrows neuroscience concepts into NAS problems, which is a good story. However, this paper has too many problems to be solved, as mentioned above. You may need lots of effort improving your paper.

---

### Decision · Program_Chairs · 2021-01-07
**Final Decision**

**Decision:**

Reject

**Comment:**

This paper would greatly benefit from some reorganization/rewriting since, as pointed out by some of the reviewers, it’s hard to follow in its current form. While a biologically inspired NAS algorithm could be an interesting direction to explore, the current paper falls short in providing evidence that the approach is well-motivated or empirically strong.  In terms of empirics, too many details are missing on the search space/architecture, ablations and comparison with existing methods. For future submissions, it would be particularly useful for the authors to explicitly discuss why they don’t find competing methods applicable to their setting.